# COVID-19, Inequality and Older People: Developing Community-Centred Interventions

**DOI:** 10.3390/ijerph18158064

**Published:** 2021-07-29

**Authors:** Christopher Phillipson, Sophie Yarker, Luciana Lang, Patty Doran, Mhorag Goff, Tine Buffel

**Affiliations:** 1School of Social Sciences, The University of Manchester, Manchester M13 9PL, UK; sophie.yarker@manchester.ac.uk (S.Y.); luciana.lang@manchester.ac.uk (L.L.); patty.doran@manchester.ac.uk (P.D.); tine.buffel@manchester.ac.uk (T.B.); 2Alliance Manchester Business School, The University of Manchester, Manchester M13 9PL, UK; mhorag.goff@manchester.ac.uk

**Keywords:** COVID-19, community participation, inequality, low income, older people, social infrastructure

## Abstract

This paper considers the basis for a ‘community-centred’ response to COVID-19. It highlights the pressures on communities weakened by austerity, growing inequalities, and cuts to social infrastructure. This paper examines the disproportionate impact of the pandemic on low-income communities, whilst highlighting the extent to which they have been excluded from debates about policies to limit the spread of COVID-19. This paper examines four approaches to assist the inclusion of neighbourhoods in strategies to tackle the pandemic: promoting community participation; recruiting advocates for those who are isolated; creating a national initiative for supporting community-centred activity; and developing policies for the long-term. This paper concludes with questions which society and communities will need to address given the potential continuation of measures to promote physical distancing.

## 1. Introduction

On 13 January 2021, it was reported that more than 100,000 people had died from coronavirus in the UK, with 1546 deaths the figure for that day alone. Approximately one in six deaths in the UK could be attributed to COVID-19 or COVID-19-related causes over the period since the start of the pandemic, with the UK having one of the worst coronavirus mortality rates in the world, at 151 per 100,000 people [1]. Much has been written about the impact of the disease, especially from a biomedical and epidemiological perspective. However, the sociological dimension is also important: pandemics are invariably viewed through particular values and belief systems; reflect economic and social inequalities within societies; and are managed through socially organised forms of care and support. Most of all, the long-term impact of pandemics is as much sociological as it is biomedical. People will almost certainly view their society in a different way during and following a pandemic: diminished in some way through the loss of partners and friends; strengthened through coming together at a time of crisis; or, conversely, weakened by feelings that some have suffered more than others.

In this paper, one aspect of the sociological dimension is explored through issues relating to neighbourhood and community, examining themes covering: social inequality, the role of social infrastructure, neighbourhood change, and community development and empowerment. The paper will argue that placing communities at the centre of responses to COVID-19 should be viewed as an essential part of managing the pandemic. Yet, as will be argued, the evidence to date indicates a lack of direct involvement of local communities in developing responses to COVID-19. This is surprising for at least two reasons: first, evidence for the ‘spatialisation’ of COVID-19, with its greatest impact being on low-income neighbourhoods, and on particular groups, such as those from minority ethnic communities. Second, because of the benefits of community engagement in responding to pandemics, especially in reaching out to groups at risk through low compliance/and or limited engagement with social media. 

In this paper, the word ‘community’ draws on the definition used by Public Health England [2]: 

‘“Community” [is a term] used as a shorthand for the relationships, bonds, identities and interests that join people together or give them a shared stake in a place, service, culture, or activity. Distinctions are often made between communities of place/geography and community of interests or identity, as strategies for engaging people may vary accordingly…community [is] an umbrella term, to cover groups of people sharing a common characteristic or affinity, such as living in a neighbourhood or being in a specific group, or sharing a common faith or set of experiences’. 

This paper examines the disproportionate impact of the pandemic on particular communities, whilst highlighting the extent to which vulnerable groups have been excluded from debates about how to limit the spread of COVID-19. It argues that the extent of inequalities revealed by the pandemic underlines the case for strengthening community-based approaches to public health. The paper focuses in particular on people 60 and older living in the UK, where nine out 10 deaths from COVID-19 have occurred, but develops arguments which are relevant to other age groups and countries. The discussion is divided into four main parts: first, an overview of the economic and social context behind COVID-19—with a particular focus on pressures affecting low-income neighbourhoods; second, a review of sociological research examining changes to community life; third, a set of proposals for building a policy for community engagement and mobilisation around limiting the impact of COVID-19; and fourth, developing a long-term community-based strategy to tackle the pandemic.

## 2. Communities under Pressure: Austerity and COVID-19

The pandemic has posed particular difficulties for many low-income neighbourhoods at a time when they had already been weakened through a combination of job losses and reduced funding from local government [3]. Christakis [4] (p. 180) makes the point that COVID-19 is not socially neutral: ‘…due to a variety of sociological and biological factors, who you are does matter. Plagues can amplify existing social divisions and often create new ones…’ Reflecting this, Build Back Fairer: The COVID-19 Marmot Review [5] (p. 24) argued that: ‘levels of deprivation and health within an area have an enormous impact on mortality rates from COVID-19, and deteriorating conditions in more deprived local areas [taking the example of England] in the years up to 2020, have meant that COVID-19 mortality has been higher than would have been the case if conditions in deprived areas had improved rather than worsened in the years leading up to the pandemic’. 

Research based on the English Longitudinal Study of Ageing (ELSA) has demonstrated a causal relationship between area deprivation and social exclusion in later life. The study revealed that older people living in deprived urban neighbourhoods had the highest levels of social exclusion compared with less deprived neighbourhoods, with evidence suggesting that this stems from barriers experienced across a range of domains including access to services and amenities, social relationships, and cultural participation [6].

Neighbourhood-based inequalities have deepened in the context of COVID-19—in the first wave of the pandemic, people (of all ages) living in the poorest parts of England and Wales were dying at twice the rate from the disease compared with those in more affluent areas [7]. There are also widening inequalities between ethnic groups, with research from the Office for National Statistics (ONS) [8] showing that, in the *first wave* of the pandemic, Black British males were 4.2-fold more likely to die from a COVID-19-related death than White British males. Bangladeshi and Pakistani males were 1.8-fold more likely to die from COVID-19 than White males, after other pre-existing factors had been accounted for, and females from those ethnic groups were 1.6-fold more likely to die from the virus than their White counterparts. Findings (for England) comparing ethnic groups between the first and second waves of the pandemic suggest fewer differences between people with a Black ethnic background and the White British group, but with the risk of death remaining substantially higher in people from Bangladeshi and Pakistani backgrounds in both waves [9].

Sze et al. [10] examined the role of ethnicity on clinical outcomes for COVID-19. Their meta-analysis of 50 studies confirmed that individuals of Black and Asian ethnicity were at increased risk of COVID-19 compared with White individuals. The authors argue that in addition to factors such as occupational risks, poor housing, and poverty, racism and structural discrimination may also play a role in increasing the risk of worse clinical outcomes. The researchers argue that within a health care context, the experience of discrimination and marginalisation: 

‘…contributes to inequities in the delivery of care, barriers to accessing care, loss of trust, and psycho-social stressors. There is evidence to suggest that ethnic minorities and migrant groups have been less likely to implement public health measures, be tested, or seek care when experiencing symptoms due to such barriers and inequities in the availability and accessibility of care, underscoring critical health disparities’ [10] (p. 12). 

The role of housing inequalities may be especially important in the context of the pandemic. The Centre for Ageing Better [11] (in association with the King’s Fund) argue that the pandemic has exposed and amplified housing-related inequalities: through the acceleration of the virus in areas of poor housing, and through measures to control the virus (such as physical distancing) which have exacerbated health problems for those restricted to their homes. Once again, minority communities have been amongst those most affected: those 55 and over from BAME backgrounds occupying homes with 30 per cent less usable space than their White counterparts.

### Social Infrastructure and Social Deprivation

The impact of COVID-19 has been further increased through cuts to what has been termed the ‘social infrastructure’ underpinning communities. Klinenberg [12] (p. 5) uses this term to refer to: ‘the physical places and organisations that shape the way people interact’. He argues that: ‘When social infrastructure is robust, it fosters contact, mutual support, and collaboration among friends and neighbours; when degraded, it inhibits social activity, leaving families and individuals to fend for themselves’ [12] (p. 5). 

Cuts to social infrastructure have been uneven in their impact across local authorities. Since 2010, according to the Marmot Review [13], the most deprived communities and places have lost more funding compared with less deprived areas. The report argues that poorer areas, where council tax receipts and business rates are already low, require a greater proportion of their funding from central government grants to local authorities, yet it is in these areas, with the greatest need, where grants have been cut the most. The Review summarised the impact on what it termed ‘ignored communities’, in the following way: 

‘Over the last 10 years, these…communities and areas have seen vital physical and community assets lost, resources and funding reduced, community and voluntary sector services decimated and public services cut, all of which have damaged health and widened inequalities. These lost assets and services compound the multiple economic and social deprivations, including high rates of persistent poverty and low income, high levels of debt, poor health and poor housing that are already faced by many residents.’ [13] (p. 94).

Limited access to green space may also restrict the ability of some groups to manage the effects of COVID-19. Lindley et al. [14] examined the health-related benefits of urban green infrastructure on different age groups. The researchers found that, in the case of older age groups, with the exception of public parks and other green areas, all other types of urban green and blue (e.g., canals) space were smaller on average in the least compared with the most affluent neighbourhoods.

The evidence suggests, then, that the pandemic entered communities which in many cases were already weakened by cuts to basic services and social infrastructure. However, the interaction between communities and COVID-19 has been additionally complicated both by responses to the disease and by changes to community life itself. It is a discussion of this aspect that forms the next section of this paper. 

## 3. Changing Communities and COVID-19

Inequality and austerity have been important factors conditioning the impact of COVID-19 [3]. However, pressures have also arisen from the way in which physical distancing measures to combat COVID-19 have been matched by a greater degree of social distancing within communities. The physical distancing associated with the pandemic—face coverings; staying two metres apart from other people outside your household; avoiding large gatherings with families and friends—seem likely to continue in some form albeit moderated over time. As a guide to social interaction, the rule on physical distancing distinguishes itself from responses to other types of crises in a significant way. As Malik [15] points out: 

‘Whilst other crises—from Aberfan to Grenfell, from Hurricane Katrina to the 2004 tsunami—have compelled people to work together to provide support and aid, Covid, and the authorities’ response to it, has required a greater individualisation of society, in which social distancing and self-isolation have become the most vital expressions of social solidarity’.

Physical distancing, as a necessary response to the pandemic, may also be said to mirror long-term changes within communities. The ONS [16], for example, in surveys measuring changes in social capital—defined as the extent and nature of our connections with others—reports declines in the period since 2011 in positive engagement with neighbours (such as exchanging favours or stopping to talk), and a decrease in the extent to which people feel they ‘belong’ to the neighbourhood in which they live. 

COVID-19 has given added emphasis to the importance of the individual’s immediate locality as a source of support and everyday contact. There is some evidence of communities coming together in the early phase of the pandemic, reflected in the weekly ‘Clap for our Carers’ which ran in the UK from the end of March to the end of May 2020. However, there are also indications that, after this initial period, this sense of unity had begun to weaken. Rutter [17], reviewing a survey of 2010 adults conducted over two time periods in March and June 2020, attributes a weakening in solidarity to factors such as perceptions that some groups were ignoring rules about social distancing; intergenerational differences—older people’s concerns with health; younger people’s worries about whether they would have jobs; and divisions around the use of technology. Rutter suggests that: ‘Some people [in the survey] felt that neighbourliness and community spirit was weaker in areas of high deprivation…as well as poverty, population churn and fear of crime were also challenges that made community-building more difficult in urban areas’.

Borkowski and Laurence [18] used data from the UK Understanding Society survey to examine trends over time in overall levels of social cohesion, as well as positive and negative changes experienced by individuals. They argued that evidence from the research literature on the effects of the pandemic (such as on financial insecurity, higher levels of stress, social isolation) could contribute to a sense of reduced cohesion within neighbourhoods—especially in the case of communities experiencing disadvantages of various kinds. The researchers concluded that: ‘…despite the positive prognoses across media/political narratives, cohesion appeared to decline quite substantially around the pandemic, compared to pre-pandemic periods. This decline occurred across all…dimensions of cohesion: both behavioural [such as] ‘talking to neighbours’…but also perceptual… such as neighbour-trust’ Borkowski and Laurence [18] (p. 15).

However, the researchers make the important observation that the negative impact of the pandemic was not shared equally across all people and places:

‘More vulnerable groups, including residents of disadvantaged communities, those with lower education, and certain ethnic minorities such as Pakistanis/Bangladeshis, ‘Other’ minorities and Blacks, all experienced a greater decline compared to their less vulnerable counterparts. For several minority groups, alongside residents of disadvantaged communities, this stronger decline had the pernicious effect of widening pre-existing inequalities with their White British and affluent area counterparts’ [18] (p. 16).

Yet, it remains the case that community responses to COVID-19 have been widespread and positive in many instances. The period since the start of the pandemic in the UK has seen the rapid expansion of mutual aid, defined as: ‘…collective co-ordination to meet each other’s needs’ [19] (p. 7), with some 3000 groups (mostly newly developed) registered over the period March to May 2020. However, Toomer-McAlpine [20] notes that this figure: ‘…does not capture the true scale of the vast network of autonomous groups working interdependently, including groups of neighbours who have set up brand new online spaces to give and get help from each other, as well as pre-existing grassroots organisations who have directed their efforts towards supporting mutual aid’. 

Reflecting these developments, the British Academy [21] (p. 68) suggests that:

‘One salient trend in community-level COVID-19 responses is the shift from local to “hyper-local” forms of intervention and organisation. Hyper-local responses, such as mutual-aid networks, often utilised digital infrastructure such as WhatsApp and Facebook groups in order to coordinate and function effectively…Digital spaces such as community Facebook groups, neighbourhood-based WhatsApp groups and local online forums… [may have become even stronger during the period of lockdown]. Crucially, effective mutual aid networks have complemented these forms of communication with physical outreach through leafleting and posters, to reach the digitally excluded’.

Despite the impressive growth in organising at a local level, the pressures associated with community organising should also be acknowledged. A report on how equalities organisations across the Greater Manchester region responded to the pandemic highlighted the speed and flexibility of activities in many cases, along with the development of new partnerships within communities [22]. At the same time, significant pressures were also noted, with the ‘huge amount of effort in responding to the crisis [unlikely] to be sustainable on a longer-term basis’. The authors of the report concluded that:

‘Because of the nature of the pandemic, and the ongoing uncertainty, this has taken its toll on organisations and staff, with a result that some, especially smaller, organisations are now struggling or in danger of being overwhelmed’ [22] (p. 27).

This is an important observation about the potential long-term difficulties facing community organisations in providing support, faced with further waves of the pandemic. In this situation, it will be essential to develop a response rooted in the networks and organisations within local communities. The next section of this paper outlines the basis for this type of approach. 

## 4. Developing a Community-Centred Approach for Tackling COVID-19

The argument of this paper is that communities have, to date, been marginalised in strategies to combat COVID-19. Christakis [4] highlights two broad ways to respond to pandemics: first, pharmaceutical interventions (PIs), such as medications and vaccinations; second, nonpharmaceutical interventions (NPIs) which can be individual (e.g., mask-wearing, self-isolating) or collective (e.g., shutting schools, banning large gatherings). 

To date, collective NPIs have largely comprised actions led by government, delivering messages, for example, through press conferences, the internet, social media platforms, and the national press. These interventions have been complemented by the work of regional and local authorities, in many cases using networks developed prior to the pandemic. However, the evidence suggests that neighbourhoods and the different groups within them have been at the receiving end of actions to combat COVID-19, rather than being treated as equal partners. As Marston et al. [23] (p. 1676) note: ‘[these actions] have largely involved government telling communities what to do, seemingly with minimal community input’. 

Absent in current NPIs is the type of community-centred model put forward by Public Health England [2] (pp. 8–9). The paper suggests that: 

‘Community (or citizen) participation, that is the active involvement of people in formal or informal activities, programmes and/or discussions to bring about planned change or improvements in community life, services and/or resources, has long been a central tenet of public health and health promotion…There is a compelling case for a shift to more people and community-centred approaches to health and wellbeing. The core concepts that underpin this shift are voice and control, leading to people having a greater say in their lives and health; equity, leading to a reduction in avoidable inequalities, and social connectedness, leading to healthier more cohesive communities’ (Authors’ emphasis).

Yet, these principles have been marginalised in the development of NPIs, notably in the type of approach from central government, with PIs, and vaccines in particular, presented as the ‘magic bullet’ for ending restrictions on social behaviour, as opposed to being integrated with neighbourhood-focused activities [3]. A number of reasons can be identified for bringing communities—as defined at the beginning of this paper—to the forefront of strategies to combat COVID-19. 

First, Marston et al. [23] (p. 1676) make the general point that: 

‘…communities, including vulnerable and marginalised groups can identify solutions: they know what knowledge and rumours are circulating; they can provide insights into stigma and structural barriers; and they are well-placed to work with others from their communities to devise collective solutions. Such community participation matters because unpopular measures risk low compliance. With communities on side, we are more likely—together—to come up with innovative, tailored solutions that meet the full range of needs of our diverse populations’.

Second, community-centred approaches are especially important in countering negative or misleading views about the effectiveness of pharmaceutical interventions. One UK poll, taken in January 2021, found that 12 per cent of those sampled said they were either unlikely or definitely would not take the vaccine (4 per cent said don’t know). (This poll is cited in Gregory, A., Wheeler, C & Shipman, T. (2021) Care home workers consider legal challenge to force their workers to take the vaccine. Sunday Times, January 17.) Of particular concern were other reports covering the first wave of the pandemic in the UK—highlighted by the Scientific Group for Emergencies [24]—that up to 72 per cent of Black people (one of the groups most at risk of COVID-19) were unlikely or very unlikely to have the vaccine. Pakistani/Bangladeshi groups were the next most hesitant ethnic group, with 42% unlikely/very unlikely to be vaccinated.

Third, targeting low-income areas with tailored public health messages is essential because of the ‘clustering’ of ‘at risk’ groups. To take two examples: areas with more overcrowded housing have also seen the worst outcomes from COVID-19. (According to Gov.UK (2020): ‘A household is overcrowded if it has fewer bedrooms than is needed to avoid undesirable sharing, based on the age, sex, and relationship of household mem-bers’. Retrieved from: https://www.ethnicity-facts-figures.service.gov.uk/housing/housing-conditions/overcrowded-households/latest.) (accessed on 22 May 2021) Of the 20 local authorities with the highest COVID-19 mortality rate, 14 have the highest percentage of households living in homes with fewer bedrooms than needed (Centre for Ageing Better, 2020). Tapper [25] highlights findings which indicate that people in some of the most deprived areas of England, including Middlesbrough, Liverpool, and the London Borough of Newham, are less likely to take a coronavirus test when they experience symptoms. In Liverpool, more than half of people in affluent areas in the south of the city were being tested during the lateral flow testing pilot scheme, but take-up in deprived areas to the north was much lower. This can be attributed to the financial penalties for low-income groups forced to self-isolate, especially those working on zero-hour contracts and similarly precarious forms of employment.

Fourth, a community-centred approach would aim to provide a complementary approach to government in terms of tailoring public health messages to particular groups and individuals. One of the weaknesses of current approaches is over-reliance on access to the internet as a means of communication. This ignores the extent of digital exclusion amongst particular groups—notably, but not exclusively, the older population. To take one example, in Greater Manchester, according to 2019 ONS figures, 57 percent of people 75 plus, and 23 per cent of those 65–74 were non-users of the internet. These age groups are likely to be further disadvantaged by the decline of local newspapers—265 closed in the UK in the period 2005–2020 [26]. Given this context, more traditional means of communication about COVID-19 is probably necessary (e.g., leaflets through doors; advertising in shops) to complement digital communication and related approaches (Greater Manchester Combined Authority have produced a booklet ‘Keeping Well this Winter’, printed copies of which have been distributed to older people across the region). 

Fifth, developing a community-centred approach is important in convincing people that their own actions really can make a difference. Christakis [4] (p. 316) makes the point that: ‘If we see pandemics purely as a function of biological details…we may be lulled into thinking there is nothing we can do to prevent or arrest such events. But if we see pandemics as sociological phenomena as well, we can more clearly recognize the role of human agency. And the more we see our own role in shaping the emergence and unfolding of pandemic diseases, the more proactive and effective our responses can be’.

Working within communities will be an essential part of developing a more proactive approach. However, in the context of physical distancing, the way we ‘practise community’ [27] is clearly different to how it might be done without the constraints imposed by COVID-19. The next section of this paper considers how a community-centred strategy might be developed, one which acknowledges the long-term impact that the pandemic is likely to have—especially for those vulnerable though age, ethnicity, or deprivation in various forms.

## 5. Community-Centred Strategies and Tackling COVID-19

This section addresses the question of developing specific strategies which can strengthen the impact of NPIs but also facilitate (where necessary) the uptake of PIs. The proposals should be viewed as a contribution to developing a new public health strategy focused on protecting low-income communities, especially in the context of measures to ‘open-up’ societies that have achieved high rates of vaccination. The focus of the discussion will be on older adults, but the examples given will be relevant to other age groups as well. The areas covered will comprise: first, promoting community participation; second, recruiting advocates for those who are isolated and/or socially excluded; third, creating a national initiative for supporting community-centred activity; fourth, developing policies for the long term.

### 5.1. Promoting Community Participation

Marston et al. [23] (p. 1767) make the point that: ‘Community participation is essential in the collective response to [COVID-19], from compliance with lockdown, to the steps that need to be taken as countries ease restrictions, to community support through volunteering’. They also emphasise ‘…the extent to which grassroots movements were central in responding to the HIV/AIDS epidemic by improving uptake of HIV testing and counselling, negotiating access to treatment, helping lower drug prices, and reducing stigma’ (p. 1767).

However, developing appropriate rules of community engagement will be complex, given any restrictions placed on large gatherings and face-to-face meetings. Despite this, incorporating direct community empowerment in constructing effective short- and long-term responses to the pandemic will be vital. Community empowerment goes much further than ‘consulting’, ‘involving’ or ‘engaging’ people. It implies a process of negotiating power and building capacities to gain access, networks and/or a voice, in order to gain more control over the decisions that shape communities. What might community empowerment mean in the context of COVID-19? Some potential areas might include:

First, drawing on methods of co-research, as developed, for example, by Blair and Minkler [28], Buffel [29] and others. Older people, trained in research skills, could play a vital role in: Deepening our understanding of attitudes towards COVID-19—especially amongst groups experiencing various forms of social exclusion;Assisting dissemination of advice and messaging about protection from the virus; andChallenging negative stereotypes of older people by emphasising the skills and knowledge which they can bring to support work to control the virus.

Second, working with ‘informal’ and ‘formal’ leaders within communities could assist the uptake of PIs and encourage people to stay as safe as possible. The importance of this has increased given evidence about misleading/false information spread through social media about vaccines in particular. On example of the importance of community leaders is the role of Imams, who in January 2021 delivered sermons in mosques across the UK which sought to reassure worshippers about the safety and legitimacy of COVID-19 vaccinations and remind them of the Islamic injunction to save lives. The move came amid evidence for anxiety within Muslim communities about the safety of vaccines, and concern about slow take-up in some parts of the UK (https://www.theguardian.com/world/2021/jan/14/imams-mosques-uk-reassure-muslim-worshippers-covid-vaccines) (accessed on 2 March 2021). The Scientific Advisory Group for Emergencies [24] (p. 7) concludes that:

‘Community engagement can identify strategies to make the vaccine more accessible, including in settings outside of formal health service provision, and increases trust between formal organisations and community members. This requires involving community leaders as partners…to promote local buy-in and develop community plans... Community forums that address the cultural and historical context of vaccine research mistreatment and including diverse representation of stakeholders can increase trust’.

Third, Gilmore [30] and colleagues suggest that COVID-19 pandemic management teams should incorporate community members into the planning, response and monitoring of standard operating procedures. They emphasise the importance of disseminating this work through the various networks within communities to ensure maximum support. Ensuring diversity in the membership of management teams is also important, especially in respect of members of minority ethnic communities, and community organisers from low-income communities.

Fourth, building on existing networks and neighbourhood organisations will be vital in developing community-based interventions. Again, this can be through both ‘informal’ and ‘formal’ networks. Gardner [31] highlights the importance of what she terms ‘natural neighbourhood networks’, these referring to the ‘web of informal relationships and interactions that enhance well-being and shape the everyday social world of older people ageing in place.’ Gardner’s research demonstrates the importance of ‘third spaces’ for older people (e.g., informal sites such as cafés, local businesses, libraries, and local streets), all of which must be considered essential sites for conveying information and supporting people during the pandemic. 

In terms of formal networks, the UK Network of Age-Friendly Communities, supported by the Centre for Ageing Better, has 40 members across the four UK countries. Many of these have taken important initiatives to support people during the pandemic, including mounting campaigns to challenge ageist narratives, developing innovative forms of social participation, and distributing information booklets targeted at older people who are not online. (For information about the work of the UK Network of Age-Friendly Communities in relation to COVID-19, see https://www.ageing-better.org.uk/age-friendly-communities-and-covid-19. Accessed on 12 February 2021). 

### 5.2. Recruiting Community Advocates

The second area for intervention concerns recruiting ‘community advocates’ for those in the community who may be unable to ensure their voices are heard, but who lack anyone who can speak on their behalf. In reality, a high proportion of older adults are able to safeguard their interests or have a ‘convoy of support’ (family, friends, neighbours) able to intercede on their behalf. However, there are increasing numbers in the population who may be vulnerable to having their interests overridden at times of crisis such as COVID-19. 

Klinenberg [32] (p. 230), in research on the impact of the 1995 Chicago heat wave, pointed to the rise of an ageing population of urban residents living alone: ‘often without proximate or reliable sources of routine contact and social support’. He pointed in particular to problems faced by older men who had outlived: ‘their social networks or become housebound and ill, often suffer[ing] from social deprivation and role displacement in their later years’.

The issue identified by Klinenberg has undoubtedly become more serious in the intervening years—with a growth in the population of men and women living alone in circumstances where accessing help may be increasingly difficult. Beach and Bamford [33], using data from the English Longitudinal Study of Ageing (ELSA), found that 14% of older men experienced moderate to high social isolation compared to 11% of women. Almost 1 in 4 older men (23%) had less than monthly contact with their children, and close to 1 in 3 (31%) had less than monthly contact with other family members. For women, these figures were 15% and 21%, respectively. The authors concluded that as the population of older men continues to grow and more people in this group find themselves living alone, social isolation and the potential issues it brings are set to get worse.

Social isolation need not necessarily be a problem if services are plentiful and easily available. However, the combination of austerity and COVID-19 has drastically rationed support of all kinds—the impact of which may be especially severe for isolated men who may, in any event, according to Beach and Bamford, be less likely to seek medical help when needed. Some of the actions taken to manage the pressures associated with COVID-19 raise particular concerns in relation to older people living alone and/socially isolated, as well as for families generally who lack confidence when dealing with the health care system. 

The Sunday Times Insight Team [34], in an analysis of admissions to hospitals during the first six months of the pandemic, found that as a result of the shortage of intensive care beds:

‘…the government, the NHS and many doctors were forced into taking controversial decisions—choosing which lives to save, which patients to treat and who to prioritise—in order to protect hospitals. In particular they took unprecedented steps to keep large numbers of elderly and frail patients out of hospital and the intensive care wards to avoid being overwhelmed… [The resulting] huge increase in [excess] deaths outside hospitals was a mixture of coronavirus cases—many of whom were never tested—and people who were not given treatment for other conditions that they would have had access in normal times. Ambulance and admission teams were told to be more selective about who should be taken into hospital, with specific instructions to exclude many elderly people. GPs were asked to identify frail patients who were left at home even if they were seriously ill with the virus’ (see, further Calvert and Arbuthnott [35]).

Those affected by the rationing of hospital care may well have been a cross-section of the older population, reflecting the diversity of social ties and circumstances characteristic of people 60 and over. However, it is also possible that some groups—such as those without families voicing concerns on their behalf—were more affected by decision making which favoured younger and fitter patients over those defined as ‘frail’. In this situation, and given the long-term pressures which health and social care are likely to experience, developing a network of advocates within communities will be important to prevent isolated individuals being denied appropriate treatment and support. Advocates could be drawn from existing organisations, for example local AgeUK branches, and Good Neighbour and Befriending groups. However, this would require resourcing to support training and financial support to those carrying out such work, an issue considered in further detail below.

### 5.3. National Funding

The third argument is for a national, government-funded initiative, to support community-centred work. Marston et al. [23] (p. 1677) make the case for funding community engagement taskforces to ensure that a community voice is incorporated into the pandemic response. They argue that this will require: ‘…dedicated staff who can help governments engage in dialogue with citizens, work to integrate the response across health and social care, and coordinate links with other sectors such as policing and education. This engagement will require additional resources to complement existing health services and public health policy. Dedicated virtual and physical spaces must be established to co-create the COVID-19 response, with different spaces tailored to the needs of different participants—e.g., different formats for discussion, timings, locations, and levels of formality’. 

Some areas may already have taskforces working along these lines, but the need both for additional funding from central government, and the importance of raising the profile of community-centred work, will be vital. This work will be especially important in developing effective policies over the longer term, given the likelihood of social distancing measures needing to continue over months or years. The implications of this last point are addressed in more detail in the final section of this paper.

### 5.4. Developing Long-Term Community-Centred Policies

The impact of COVID-19 can be measured in a variety of ways—in terms of quality of life, lost income, mortality, and long-term illness. Reflecting on all these, we know that the pandemic has already accelerated the decline in life expectancy which had started to affected poorer areas in England and Wales over the period 2010–2020: Aburto and colleagues [36] estimate a fall of about one year since the start of the pandemic. The hope of course is that the pandemic will be of (relatively) short duration, with access to vaccines able to stem the tide of deaths and sickness. Yet, this seems unlikely and for a variety of reasons. Christakis [4] makes the point that COVID-19 needs to be placed within the wider context of globalisation, mass migrations, and increased urbanisation, these forces contributing to the persistence of infectious diseases. He argues that:

‘Outbreaks of novel pathogens reflect, among other things, changes in the way in the way humans come into contact with animals. In fact, two of the biggest challenges humans face—extreme weather events…and periodic outbreaks of serious diseases—may be linked by climate change. People driven from their homes by changes in the weather or people clearing new land for cultivation may come into contact with animals (who may also be driven from their homes) in ways that increase the likelihood of the emergence of new pathogens in our species’ (pp. 298–299).

However, it might be argued as well that increased instability in the world coincides with the rise in populations (such as those comprising people over 60) who are especially vulnerable to infectious diseases. COVID-19 (or some variant) is, however, likely to persist for some time for many other reasons [3]. PIs—for those countries that can afford them—will certainly be vital in controlling the spread of the virus. At the same time, as many commentators have pointed out, many ‘unknowns’ remain: their affordability (for many countries), their efficacy against new mutations, and their supply. 

Given this context, developing neighbourhood-level public health systems will be essential to run alongside successive programmes of vaccinations. Developing this argument, three priorities might be highlighted: 

First, community-centred work needs to be placed within a wider context of ‘community development’. COVID-19 has preyed on neighbourhoods damaged by cuts to basic services and social infrastructure, lack of investment in housing, and the rise of precarious forms of employment. Any long-term strategy to combat the pandemic has to be rooted in addressing the multiple forms of deprivation affecting many communities in the UK. These, as the evidence shows, are drivers for transmission of the virus, notably through overcrowded households, with household members employed in high-risk occupations passing the virus across generations [37]. 

However, community development must also come from ‘below’, with the pandemic giving impetus to what Sennett [38] (p. 143) refers to as ‘localised sociability’, assisted by the strengthening of neighbourhood-based organisations. This may be especially important given (at the time of writing) the impact of three lockdowns (in the case of the UK) on potentially reinforcing social isolation amongst some groups. The effect of successive lockdowns remains unclear: for example, in creating a loss of confidence in moving around neighbourhoods; re-establishing relationships; and developing new contacts. One possible consequence will be the need to establish new forms of solidarity within communities, drawing on the collective organisation of older people. Relevant examples which emerged before the pandemic include the ‘Village’ movement, and Naturally Occurring Retirement Communities (both developed in the (USA), and the growth of the World Health Organisation’s global network of age-friendly cities and communities [39]. These, and other approaches, provide useful models for the direct involvement of older people in re-building communities in which they are likely to have spent a significant part of their adult life. 

Second, COVID-19, as numerous reports have made clear, has exposed and exacerbated longstanding inequalities affecting BAME groups in the UK. Racism and discrimination also play an important role in this regard, as highlighted in the research by Sze and colleagues [10] cited earlier in this paper. However, much of this was predictable given available knowledge about poverty, co-morbidities, poor-quality housing, and low incomes, affecting many of those in South Asian, and other BAME communities. The question is why there was a failure to develop preventative forms of community-centred working with BAME groups from the beginning of the pandemic. Such targeted work, involving community leaders wherever possible, will certainly be essential over the medium and longer term. However, as suggested earlier, this type of initiative will require additional sources of funding to support what are financially constrained organisations even in ‘normal times’. 

Third, COVID-19 has proved catastrophic for people in residential care—in the UK as well as for many other countries. By mid-January 2021 in the UK, one-third of fatalities had been care home residents—32,000 people after taking into account residents who had died after being admitted to hospital [40]. This is an extraordinary figure, which indicates a *systemic* failure to safeguard a highly vulnerable group. 

Bold thinking is certainly needed by the research and policy community about the future of residential and nursing home care: challenging rather than colluding with current models of care. Privatisation has proved a flawed model; but the public or not-for-profit sector does not provide a straightforward solution either. The way forward must certainly be to ‘downsize’ from ‘industrial-scale’ care, potentially looking at placing the management of homes within a local authority framework. Crucially, such homes should be embedded in their surrounding neighbourhood. Developing viable models which provide some degree of protection for people will be challenging, but the impact of COVID-19 has confirmed the urgent need for major reforms of the residential and nursing home sector. 

## 6. Conclusions

In the concluding chapter of his book on the impact of the pandemic, Christakis [4] (p.318) explores the question of ‘How Plagues End’. He suggests that social variables and values play an important role in thinking about for whom a pandemic has ended: ‘For the elderly, chronically ill, the poor, the imprisoned, and the socially marginalised, the SARS-2 pandemic might continue to be a threat biologically long after the majority of the population has moved on psychologically and practically and long after overall levels of the virus are low’. 

The importance of this point is now even clearer in the period that has elapsed since the start of the pandemic. For low-income countries, there may be no obvious end in sight—especially for those countries who either cannot afford the vaccine or whose health care systems will find its storage and distribution difficult to manage. Modelling undertaken by the Economist suggests that the pandemic is increasingly (as at Spring 2021) concentrated in developing countries. Accordingly: ‘Death rates among poor young populations are much higher than they would be for countries in the rich world with similar age profiles. And for the elderly in poor countries the outlook is grim. South Africa has seen 120,000 excess deaths among those over 60’. (Economist, 15 May 2021: 18). 

For high-income countries, affordability and management may be less of an issue, but the threat of mutations and the need for annual (at least) booster injections will maintain the pressures associated with the pandemic for some time to come. In the UK, the decision to ‘open up’ society (in the summer of 2021) was made notwithstanding the 18.5 million people (24.4% of the UK population) at increased risk of developing severe COVID-19, including people with underlying health conditions, minority ethnic groups, and older people [41].

As this last point would suggest, complex issues are being raised about the balance between removing restrictions on social behaviour, whilst protecting those most vulnerable to the effects of the pandemic. Some important questions for public policy include: will societies continue to support measures which hold back their economies in the interests of groups such as older people—who continue to suffer the bulk of fatalities associated with COVID-19 (people 80 and over diagnosed with the disease are 70-fold more likely to die than those under 40)? Will special forms of protection be necessary to support those for whom the pandemic may be never-ending: people in residential care (unless the sector is drastically reformed); people on the margins of society (e.g., the homeless); people forced to continue working despite carrying the virus (e.g., those on zero-hour contracts); and those living in overcrowded houses (e.g., multi-generational families in particular)? These questions make the case for a new public health policy to address the needs of a society which has the aspiration to be post-pandemic, but which is likely to be forced to move relatively slowly towards that goal. Supporting this task must be the knowledge gained in working with communities to assist those most affected by the virus, and drawing on this experience to tackle the economic and social inequalities which have been part of the social and biological construction of COVID-19.

## Data Availability

Not applicable.

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
