# Peer review of "COVID-19, Inequality and Older People: Developing Community-Centred Interventions"

_ijerph, 2021, doi:10.3390/ijerph18158064_

Round 1
Reviewer 1 Report
The article considers the basis for a ‘community-centred’ response to COVID-19 and shows that the end of the pandemic will not be the same for all people. The manuscript includes many publications that show that there is a sociological dimension of the Covid-19 pandemic that goes beyond the biomedical consequences, that must be taken into consideration, because the sociological consequences will persist after the pandemic. Congratulations for the authors for the interest in assisting the inclusion of the neighbourhood and community in strategies to tackle the pandemic.
The manuscript is well structured and describe the economic and social context behind COVID-19, with a sociological research examining changes to community life. What is more, authors formulate strategic proposals to strengthen community engagement to tackle the pandemic. The conclusions are interesting for the readership of the Journal and will attract a wide readership now that the biomedical dimensions of the pandemic are being left behind, but the sociological dimension will remain for a long time. Publishing the article will provide an advance towards the current knowledge in terms of tools to go out of the crisis caused by the pandemic without leaving anyone behind.
The article written in an appropriate way, and the bibliographic citations are correct.
Author Response
We appreciate the reviewers comments about the value of a sociological analysis focused on neighbourhood and community, and the importance of developing public health tools as we move out of the pandemic. The revised paper tries to strengthen the key steps in the arguments relating to this theme.
Reviewer 2 Report
This is a timely and informative paper, advocating for a re-articulation of community, as well as tackling associated inequality, in coping with pandemic. The paper is well-written and the flow of logic is quite clear. I support publication of this paper. I have some minor comments for authors to consider, where appropriate.
1. In the introduction (also conclusion), perhaps the author(s) can highlight their foci for a new public health policy as "COVID 19 is not socially neutral", which requires reconceptualising our public health or introducing "new" public health. In this way, the discussions under different sections can be better framed under this theme.
I quite like the argument that the COVID 19 urges people to think alternative ways to promote solidarity. For example, in page 4: "... social distancing and self-isolation have become the most vital expressions of social solidarity". Perhaps the authors can elaborate, under community approach or strategy section, some possible community development strategies/paradigms that can respond to such changes.
In the conclusion part, in terms of a new public health policy, can author(s) highlight some key implications for policymaking in post-COVID era? Besides, "...For the elderly, chronically ill, the poor, the imprisoned, and the socially marginalized, the SARS-2 pandemic might continue to be a threat biologically long after..." - in what way and how public policy can respond?
Apart from different social groups, locations/geography determine various degrees of access to health related resources. This includes presence of green, open space and easy access to daily necessities during COVID19. Under "communities under pressure: austerity and Covid-19", perhaps author(s) can consider incorporating some discussions about spatial factors to supplement arguments under neighbourhood-based inequalities.
Author Response
We appreciate the very helpful suggestions from Reviewer 2.
Comment 1: We have given more prominence to the issue of framing the article around the need for a 'new public health' approach to the pandemic (e.g. pages 2 and 7 and in the revised conclusion).
Comment 2: The suggestion of providing examples relating to social solidarity was a valuable one: we have indicated some approaches on page 11.
Comment 3: The conclusion has been modified with some questions for public policy which follow a clarification about the issue of policymaking in the post-pandemic era.
Reviewer 3 Report
- This paper is more like a report than an academic journal article. The research methods are lacked in this paper and most of the contexts in the paper are from literature review. Also, there is no discussion of the academic definition of community and only applied definition from Public Health England. There is also lack of a literature review of community development theory and practice,
- This article argued that the COVID-19 pandemic has changed the community resulting in isolation and weaker the community spirit due to the physical distancing and lockdown. But the proposed solution of developing a "community-centred approach" or NPI, failed to explain how will it work under the lockdown situation? I would suggest the authors to clarify this point in detail.
- A lot of terms applied in the article need accurate definitions, such as, "overcrowded houses", "community renewal".
- Four community-centred strategies proposed in the article (P.7-P.9) are the common approaches that have already applied in most community development projects worldwide including community empowerment, promoting participation, working with informal and formal leaders....for a long time. The authors need to explain what are the differences between the strategies proposed in the article and the existing approaches. You need to point out what existing strategies do not work under the pandemic and what will work after your research.
- Older people are only one of the vulnerable cohorts might encounter inequality situation. It is weird to have it in the title since only two community-centred strategies mentions about older people.
Author Response
We appreciate these valuable comments from Reviewer 3. Our responses are as follows:
Comment 1. Yes the issue of green, open space is especially important and we provide a new reference to inequalities on this aspect on page 4. We do reference the importance of spatial aspects in the paper (e.g. page 2), and the work of Michael Marmot is especially important in this regard (page 2.). We are not sure that we see the paper as a 'Report'; rather than as an attempt to provide a distinctive sociological framework and argument for developing an alternative public health approach for coming out of the pandemic - around which our use (rather than 'review' of the research literature) is based. Given the aim of the paper, we think our focus on a 'public health' definition of community was justified: a review of the broader literature on community development theory and practice was beyond the scope of the paper.
2. The reviewer raises the important point of how our approach would work under lockdown conditions: in the revised paper we have clarified this (e.g. page 7) to indicate that we are increasingly looking at the methods suggested in the context of societies which are loosening restrictions on social behaviours.
3. Reviewer 3 queries some of the terms used in the paper, in particular 'overcrowded housing' and 'community renewal'. On the former, we have now given the official definition from the Gov.UK website; on the latter, we agree the term 'renewal' is ambiguous and have substituted 'development' instead.
4. Comment 4 raises the issue of what 'strategies do not work' under the pandemic and 'what will work after our research'. On the former, our argument (.e.g pages 5-7) is that the existing reliance on pharmaceutical interventions and top-down non-pharmaceutical approaches) has been a mistake, for reasons we give. On the latter, we set out a series of proposals which will of course need to be tested in the field and evaluated accordingly.
5. We agree that 'older people are only one of the vulnerable cohorts that might experience inequality'. We stress on pages 2 and 7 that although our focus is on older people, we do think our arguments are applicable to other age groups. Of course much of the research we examine is skewed towards that of older age groups.